# Two Potential Clinical Applications of Origami-Based Paper Devices

**DOI:** 10.3390/diagnostics9040203

**Published:** 2019-11-26

**Authors:** Zong-Keng Kuo, Tsui-Hsuan Chang, Yu-Shin Chen, Chao-Min Cheng, Chia-Ying Tsai

**Affiliations:** 1Institute of Nanoengineering and Microsystems, National Tsing Hua University, Hsinchu 30013, Taiwan; ayudivac@gmail.com (Z.-K.K.); ericys2004@gmail.com (Y.-S.C.); 2Institute of Biomedical Engineering, National Tsing Hua University, Hsinchu 30013, Taiwan; allmyown74@gmail.com; 3Department of Ophthalmology, Fu Jen Catholic University Hostpital, Fu Jen Catholic University, New Taipei City 24352, Taiwan; 4School of Medicine, College of Medicine, Fu Jen Catholic University, New Taipei City 24205, Taiwan

**Keywords:** origami-based paper analytic device, origami ELISA, IgG, paraquat

## Abstract

Detecting small amounts of analyte in clinical practice is challenging because of deficiencies in specimen sample availability and unsuitable sampling environments that prevent reliable sampling. Paper-based analytical devices (PADs) have successfully been used to detect ultralow amounts of analyte, and origami-based PADs (O-PADs) offer advantages that may boost the overall potential of PADs in general. In this study, we investigated two potential clinical applications for O-PADs. The first O-PAD we investigated was an origami-based enzyme-linked immunosorbent assay (ELISA) system designed to detect different concentrations of rabbit IgG. This device was designed with four wing structures, each of which acted as a reagent loading zone for pre-loading ELISA reagents, and a central test sample loading zone. Because this device has a low limit of detection (LOD), it may be suitable for detecting IgG levels in tears from patients with a suspected viral infection (such as herpes simplex virus (HSV)). The second O-PAD we investigated was designed to detect paraquat levels to determine potential poisoning. To use this device, we sequentially folded each of two separate reagent zones, one preloaded with NaOH and one preloaded with ascorbic acid (AA), over the central test zone, and added 8 µL of sample that then flowed through each reagent zone and onto the central test zone. The device was then unfolded to read the results on the test zone. The three folded layers of paper provided a moist environment not achievable with conventional paper-based ELISA. Both O-PADs were convenient to use because reagents were preloaded, and results could be observed and analyzed with image analysis software. O-PADs expand the testing capacity of simpler PADs while leveraging their characteristic advantages of convenience, cost, and ease of use, particularly for point-of-care diagnosis.

## 1. Introduction

To improve the operation and expand the testing capacity and scope of paper-based analytical devices, we borrowed from the art of origami and folded papers into functional forms that facilitated the application of multiple reagents for conducting more complex and potentially more impactful paper-based, point-of-care biochemical analyses, including multiple, simultaneous chemical reaction-based assays and enzyme-linked immunosorbent assays (ELISAs). In recent years, paper-based analytical devices have demonstrated a variety of advantages for point-of-care diagnostics. Such devices are inexpensive, easily obtained, ecofriendly, naturally wicking, highly compatible with bioassays, and require only small sample amounts [1,2]. Accordingly, a wide array of bioassays have been developed using paper-based analytical devices, including ELISA, and commercialized rapid tests for influenza, bacterial infection, and pregnancy. For example, paper-based ELISA (P-ELISA) devices have been applied to diagnose biological sample protein targets, such as HIV [3], VEGF in aqueous humor [4,5,6], lactoferrin in tears [7], autoimmune antibodies in serum and blister fluid [8], human chorionic gonadotropin (hCG) in urine samples [9], a cancer marker (prostate-specific antigen, PSA) in serum [10], and *Escherichia coli* in water [11]. The list of analytes identifiable in urine or serum samples using paper-based analytical devices and biochemical analysis includes proteins, glucose, lactate, uric acid, pesticides, and others [4,12,13]. Although paper-based analytical devices are very competitive in terms of cost and sample requirements, they may yet be improved upon. Some P-ELISA procedures, for example, require a number of different reagents and complicated procedures that can be disadvantageous for point-of-care (POC) testing. Further, a more critical environment for biochemical reactions may be required for colorimetric detection of several analytes such as paraquat, a poisonous organophosphate [4,13], and accuracy may suffer when using small sample amounts in paper-based analytic devices. Here, we investigate the possibilities of a user-friendly origami-based paper device to ameliorate P-ELISA complexity and provide a more suitable environment for biochemical analysis.

Keratitis is a leading cause of ocular blindness globally. Gram-positive and gram-negative bacteria, viruses such as herpes simplex virus (HSV), and parasites such as *Acanthamoeba* are known to cause keratitis. Diagnosing keratitis from nonbacterial pathogens, i.e., from HSV or *Acanthamoeba*, is difficult, relies on clinical symptoms and signs, and requires specific treatment such as antiviral agents or chlorhexidine [14,15]. Early and specific diagnosis of keratitis and its cause is important for prognosis [16]. This may be more easily achieved with new methodology. Tears provide the first line of immunological defense for the ocular surface, and tears contain many proteins and cytokines that might be measured as markers of the local immunological state. Secretory immunoglobulins in tears, e.g., IgA, IgG, and IgM, are thought to be specific antimicrobial substances that may increase following ocular surface infection [17]. Secretory IgA, in particular, has been shown to protect the ocular surface from viral and bacterial infection, as well as from parasite infestation, and may operate by coating pathogenic microorganisms to prevent them from adhering to the corneal epithelium [18,19]. IgG and IgM are present in very low concentrations in tears, and IgG concentration is known to increase in inflammation [17,18,19,20]. In HSV infection, IgG could sensitize HSV, and lead to increased viral load [21,22] and secretory IgA may neutralize HSV [23]. The effect of neutralization of HSV in ocular tissue may depend on the concentration ratios of IgA and IgG [23]. One study showed similar total serum IgG level but significantly higher anti-*Acanthamoeba* IgG antibody levels in patients with *Acanthamoeba* keratitis compared to those in normal subjects [24]. Patients with *Acanthamoeba* keratitis also displayed lower levels of IgA in tears compared to normal subjects [24]. Efficient methods to detect levels and ratios of each of these compounds would be useful for diagnosis and treatment.

Paraquat is a commonly used herbicide around the world due to its low cost and ready accessibility, especially in developing countries [25]. It has been prohibited in many countries due to its lethal toxicity, but in many developing countries, paraquat is still widely used. Lethally exposed patients often die from multiple organ failure, and without adequate antidote, the mortality rate from exposure is as high as 60–80% [26,27,28]. In one study, repeat pulse therapy within 5 h of paraquat ingestion decreased the mortality rate to less than 42.9% [29]. To achieve treatment within 5 h, rapid diagnosis of paraquat is invaluable. A paper-based device has been developed to detect paraquat in human serum and could be used in developing countries [6,13]. In the process of detecting paraquat, it is very important to keep the samples wet to ensure accuracy, but this is difficult to do with the existing paper-based device. A folded device that protects the testing zone from drying out may be more useful for maintaining a moist environment that would be more conducive to accurate testing.

Origami is the ancient art of folding flat paper to fabricate three-dimensional sculptures or structures [13,30]. This ancient art form has recently been used with new, scientific intent, as several origami-influenced, paper-based analytical devices (O-PADs) have been developed by folding a single sheet of flat paper into elegantly functional designs using a single patterning step. This method eliminated the need for complicated, sequential, layer-by-layer stacking of individual layers of paper using double-sided tape [31]. In addition, these microfluidic O-PADs can be unfolded to reveal each layer for easy test result analysis [31]. The main purpose for leveraging origami in these studies was to simplify the fabrication of three-dimensional (3D) microfluidic channels or multiple working zones within paper microfluidic devices. Despite some design advances, these devices could not ameliorate the need for multiple reagent preloading or provide the optimal environment for detection [32,33]. Several studies have been undertaken to investigate the possibilities of using origami to fabricate PADs for performing bioassays [31,32,33,34,35], but each did require the use of multiple reagents. In this manuscript we describe our research into possible approaches for optimizing O-PADs that could be useful for continued research efforts.

We explored two particular potential clinical opportunities for POC O-PADS to demonstrate their advantages: (1) The development of an O-PAD ELISA for IgG level testing that reduced test complexity and the number of required reagents; (2) the development of an O-PAD for biochemical analysis that used paraquat as the proof-of-concept assay target. We believe our work could lay some groundwork for the use of O-PADs as an improvement to PADs for POC testing.

## 2. Materials and Methods

### 2.1. Design of Origami-Based PAD for ELISA

The O-PAD for ELISA (O-ELISA) was designed on Whatman qualitative filter paper No. 1 and patterned using a wax printer (Xerox Phaser 8650N color printer, Norwalk, CT, USA). Four wings were fashioned to house ELISA reagent loading zones for pre-loading, and the central area was designed and reserved as a testing zone for sample loading. After heating at 105 °C for 3 min, the melted wax wicked through the paper to create hydrophobic barrier wells in the paper. Reagents were then pre-loaded into reagent zone barrier wells. These reagents included the following: (1) 2 µL of blocking buffer (5% (*w*/*v*) bovine serum albumin (BSA) (Sigma, SI-A7906, St. Louis, MO, USA) in PBS (Corning, 21-040-CM, Corning, NY, USA); (2) 2 µL solution of alkaline phosphatase (ALP)-conjugated detection antibody (Cell signaling, #7054, Danvers, MA, USA) (20 µg/mL, 0.01% (*v*/*v*) Tween-20 (Sigma, P9416, St. Louis, MO, USA); and (3) 2 µL BCIP/NBT substrate (13.4 mM BCIP (Sigma, B6274), 9 mM NBT (Sigma, N5514), 25 mM MgCl_2_ (Sigma, M8266), 500 mM NaCl (Sigma, S7653) in 500 mM Tris buffer pH 9.5 (Sigma, T4661). After loading each reagent, the device was placed under ambient conditions for 5 to 10 min until the reagents dried.

### 2.2. Performing Origami-Based PAD for ELISA

The process for creating and using O-PADS is outlined in Figure 2. Briefly, we first loaded 2 µL of sample into the testing zone and allowed it to dry for 10 min under ambient conditions. Then, we folded the reagent zone to contact the testing zone, and loaded 3 µL of PBS into the reagent zone and allowed PBS to transfer reagents to the testing zone. The reaction time for BSA blocking was 5 min and the time for reaction with ALP-conjugated antibody was 10 min. Then, we folded the wash zone to the testing zone, placed the O-PAD on paper towels and washed with 3 µL of PBS. Finally, we folded the substrate zone to the testing zone and loaded 3 µL of PBS to transfer substrate to the testing zone and allowed the enzymatic reaction to proceed for 20 min under ambient conditions. The O-PAD was subsequently scanned using a photo scanner (EPSON 3490 PHOTO). Gray-scale color intensity following the enzymatic reaction was quantified using ImageJ software (ImageJ 1.80, National Institutes of Health, Bethesda, MD, USA). In order to diminish both background noise and analytical variation, we subtracted the original background value of each test zone for each detection result. The experimental data were further analyzed using Sigmaplot (version 13.0, Systat Software, Inc., San Jose, CA, USA).

### 2.3. Fabrication of the Origami-Based PAD for Paraquat Detection

An O-PAD for paraquat detection is shown in Figure 3. The pattern created on the device was fabricated as described above. We preloaded 5 µL of 5N NaOH and 6 µL of 5% (*w*/*v*) ascorbic acid onto the left and right parts of the PAD, respectively. After loading each reagent, the device was placed under ambient conditions for 20 min until the reagents dried.

### 2.4. Performing Origami-Based Detection of Paraquat

To perform origami-based detection of paraquat, we sequentially folded the left and right parts of the structure over and onto the central portion. We then loaded 8 µL of sample onto the test zone and allowed it to rest for 10 min at 25 °C. Detection results were recorded using a digital camera and the signal was analyzed using ImageJ software. The RGB color value of each detection result was obtained, and the R value was used for further calculation [13,36]. We subtracted original test zone background values for each detection result.

## 3. Results and Discussion

### 3.1. O-PAD for ELISA

We designed and fabricated a P-ELISA device with multiple preloaded reagents and, borrowing from origami, folded it to make a multilayered, three-dimensional device that provided structural advantages. The design of this O-PAD device is provided in Figure 1a,b.

We used rabbit IgG as the analyte for demonstrating device performance (see schematic illustration in Figure 2. We used serial dilutions of IgG diluted in PBS (260 nM–2.6 pM) to investigate the limit of detection (LOD). The total time to complete testing with this novel device was approximately 46 min. Results from our test were scanned using a photo scanner and the mean gray-scale intensity of the color formed was measured using ImageJ (Figure 3). The LOD for this rabbit IgG system was calculated using the generated colorimetric results. We graphed colorimetric intensity versus rabbit IgG concentration (log scale) to produce the standard curve of this rabbit IgG system. We further employed nonlinear regression using the Hill equation, to generate a sigmoidal curve fit that has previously been used to describe the antibody–antigen reaction kinetics in paper [3]. Similar sigmoidal fits for both rabbit IgG [3] and human vascular endothelial growth factor [6] paper-based ELISA studies have been demonstrated. In the Hill equation (Equation (1)), è represents the fraction of occupied binding sites, [L] represents the ligand concentration (in mol), [L_50_] describes the ligand concentration when half of the binding sites are occupied (in mol), and n is the Hill coefficient. The fraction of occupied binding sites can be described by the ratio of the observed intensity (I) of the colorimetrical signal to the maximum intensity (Equation (2)). The intensity is proportional to the amount of detected antigens (Equation (3)).
è = [L]^n^/([L]^n^ + [L_50_]^n^)(1)
è = *I*/*I*_max_(2)
*I* = *I*_max_[L]^n^/([L]^n^ + [L_50_]^n^)(3)

Accordingly, we fitted the generated colorimetric results into the standard curve using Sigmaplot, and determined that the LOD of the IgG system using our device was about 201 pM, which was calculated by fitting the signal (25.4) that was three times the standard deviation of the blank control [3]. This low IgG LOD could be beneficial for analysis of low-volume specimens in clinical practice. Tear fluid for instance, an example of a low-volume sampling source, demonstrates increased inflammation caused by giant papillary conjunctivitis with an IgG concentration of 160 nM (i.e., IgG detection limit of 160 nM) [37] and viral infection, such as HSV [38], and could benefit from the availability of such a potential diagnostic tool.

Origami has previously been applied to fabricate a paper-based device for performing bioassays [32,33,39]. While the process requires the use of multiple reagents and complex steps, end-user time is saved by preloading with the necessary reagents for performing ELISA. With this ELISA O-PAD, users only need to load their sample to the test zone and apply a single, fixed volume of buffer to initiate testing. Results can be easily obtained via colorimetric readout. Moreover, the colorimetric signal can be scanned or photographed using a smartphone and the results more finely analyzed and compared. The mean color formation intensity can be quantified using versatile software applications including ImageJ or APPs, which can provide convenient POC diagnostic results.

Although an O-PAD with pre-loaded reagents is convenient for POC testing, the storage requirements of proteins, enzymes, substrates, and reagents may limit practicality. We used alkaline phosphatase-conjugated antibody as the detection antibody because commercialized alkaline phosphatase was provided to us as lyophilized powder shipped at ambient, room temperature. BCIP/NBT, the colorimetric substrate of alkaline phosphatase, was available in tablet form and could be crushed to a powder. For both of the above reasons, we felt that a preloaded alkaline phosphatase/BCIP/NBT system was suitable for our O-PAD. The horseradish peroxidase (HRP) and 3,3′,5,5′-tetramentylbenzidine (TMB) system commonly used for ELISA kits may be unsuitable for O-PAD development because the hydrogen peroxide involved in the colorimetric reaction is unstable [40], but we noted that Ramachandran et al. reported a method for long-term dry storage of HRP-conjugated antibody as well as its colorimetric substrate, diaminobenzidine (DAB) [41]. Indeed, some enzymes or proteins that have not undergone optimized drying procedures in suitable buffers may not be stable in a dried state for long-term storage. Our manuscript describes a process to leverage the known advantages of P-ELISA in a novel manner by folding it, origami-fashion, and pre-loading multiple reagents to create a multiplexed diagnostic tool. We believe that the versatility of this approach could lead to an exceptionally useful commercial product, especially if long-term dry-state storage solutions can be found for a broad array of test reagents.

A three-dimensional (3D) microfluidic paper-based analytical device (3D-μPAD) was reported by Liu et al. that used a sliding movable test strip that was dipped into reagent stored within the device [37]. After sliding the test strip to a different location under spots preloaded with stored reagent, buffer was loaded and the integrated, stored reagent was transferred to the test zone. ELISA could be completed by repeating this test-strip sliding action and loading additional buffer [37]. Using rabbit IgG as a model analyte, Liu et al. performed an ELISA in 43 min, with a detection limit of 330 pM. Although this device provides an alternative solution to previous P-ELISA methods, it was difficult to control the contact between test zones and stored, preloaded reagents. Moreover, mass fabrication of this particular 3D-μPAD was problematic.

### 3.2. O-PAD for Paraquat Detection

Our previous results showed that a folded, origami-inspired structure could facilitate efficient PAD procedures. However, successful detection of multiple analytes using a single PAD relied on maintaining particular, sometimes critical, conditions such as moisture level. In a previous study, we showed that paraquat detection in a PAD required a wet environment [13,36]. Traditional ELISA, however, requires application of multiple reagents, and the time to complete such a process using paper increases the likelihood of any moisture-sensitive components drying out. To remedy this we created an O-PAD for paraquat detection that used its protective folds to maintain moisture during the sample testing phase. This O-PAD for paraquat detection, with a printed wax pattern created as described above, is shown in Figure 4. We preloaded 5 µL of 5 N NaOH and 6 µL of 5% (*w*/*v*) ascorbic acid (AA) onto the left and right parts of our PAD, respectively. After preloading each reagent, we placed the device under ambient conditions for 20 min to allow the reagents to dry. We then folded the reagent zones containing NaOH and AA onto the test zone sequentially and loaded 8 µL of test sample onto the test zone before letting the device sit for 10 min at 25 °C. Paraquat, in contact with NaOH and AA, changed to paraquat radical ion (blue color) [13,36]. The resulting colorimetric response for each reagent was recorded using a digital camera and analyzed using ImageJ software. The RGB color value of each detection result was obtained. Because previous studies indicated that R value provided the best colorimetric reaction for evaluative and diagnostic purposes [13,36], we employed the same protocol here. In order to diminish background noise and analytical variation, we subtracted any original background values for each test zone result. We used five different concentrations of paraquat standard solution (i.e., 0, 10, 25, 50, and 100 ppm) to establish our standard curve (Figure 5). The LOD for paraquat detection using our O-PAD was 5.82 ppm, which was calculated by using the signal results of our 0 ppm groups plus three-fold standard deviations in the regression model of the standard curve (Figure 5).

The LOD (5.82 ppm) using our O-PAD was comparable to the LOD (3.86 ppm) of a previous 96-well format PAD we developed [13]. The O-PAD was more convenient for users because the relevant reagents were preloaded and the results could be easily obtained following a single test sample loading step. Further, the O-PAD comprising three folded paper layers provided a moist environment for paraquat testing that was more suitable than the 96-well format testing device, which consisted of a single layer of paper. These advantages could provide useful point-of-care device development strategies for other analyte detection models. It is worth noting that the lack of long-term storage stability for preloaded reagents may limit the use of our O-PAD for paraquat detection, i.e., ascorbic acid oxidizes in the presence of oxygen. We believe that methods aimed at improving long-term, dry-state reagent storage could help to optimize and broaden the range of potential applications for O-PADs.

Recently, PADs have demonstrated advantages for POC diagnostics. Paper-based ELISA, for instance, require less sample volume and less time to complete than conventional ELISA because of the high surface-to-volume ratio of paper fibers within the test zones [3]. However, PADs are still plagued by some shortcomings that remain to be resolved (Table 1). For example, multiple reagents and more consumables are required to conduct assays, which increase process complexity. Moreover, single-layer PADs may suffer from dry conditions that are less ideal, even insufficient, for testing particular analytes. To ameliorate these disadvantages, we sought to leverage the folded structure of origami to fabricate user-friendly, sensitive, and accurate PADs. Reagents required for conducting assays on such a device could be preloaded and integrated among the device’s protective folds. Use of one of these devices requires relatively straightforward folding and unfolding manipulations. Together, these advantages could increase ease, convenience, and acceptability of such devices especially for POC applications in resource-limited areas. We believe that reliance on an ancient art form, origami, could expand the future potential for the development of PADs for clinical diagnostics that could reap real benefits for human health.

## 4. Conclusions

O-PADs provide a more convenient, reliable, and easy-to-use approach for target sample detection. In this study, we successfully used an O-PAD to detect rabbit IgG level with a low LOD. Due to the limited information regarding the IgG LOD for ophthalmic viral infection or inflammation, we could not compare our results to the references. Overall, this study demonstrated two potential areas for clinical application of OPADs that could provide significant scientific and health benefits. Our device provides two advantages: (1) to reduce test complexity and the number of required reagents through pre-loaded reagents; and, (2) to provide a wet environment for moisture-sensitive analysis, e.g., paraquat detection. However, there are still several limitations associated with our device. First, while providing advantages over paper-based ELISA, our device still could not reach the sensitivity of conventional ELISA; this limitation may be improved by optimizing the assay to suppress the background signal. Second, the long-term storage stability of preloaded reagents such as ascorbic acid, which oxidizes if exposed to oxygen, may limit usage. We believe that additional developments, especially those aimed at improving long-term dry-state storage, would improve our device and broaden its applicability.

## Figures and Tables

**Figure 1 diagnostics-09-00203-f001:**
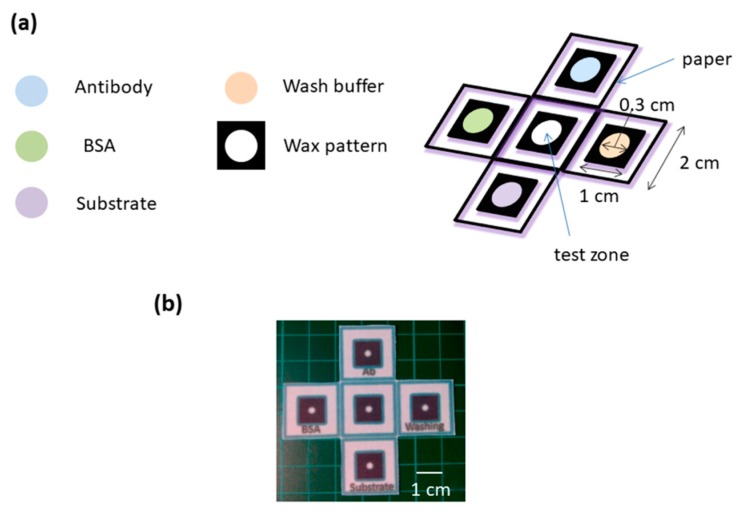
The design of an origami-influenced paper-based analytic device (O-PAD) for enzyme-linked immunosorbent assay (ELISA). (**a**) Reagents were pre-loaded into the four-wing fashioned zones, and the central area was reserved as testing zone for sample loading. (**b**) The picture of our O-PAD.

**Figure 2 diagnostics-09-00203-f002:**
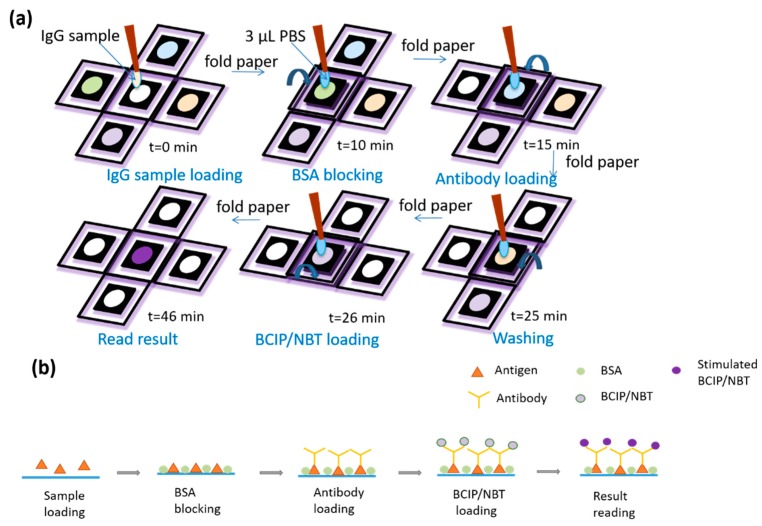
Schematic illustrations demonstrating use of O-PAD ELISA. (**a**) Reagents were pre-loaded into reagent zone barrier wells. These reagents included the following: (1) 2 µL of BSA blocking buffer (2) 2 µL solution of alkaline phosphatase (ALP)-conjugated detection antibody (3) 2 µL BCIP/NBT substrate. After loading each reagent, the device was placed under ambient conditions for 5 to 10 min until the reagents dried. (**b**) Schematic illustrations for BSA blocking, antibody loading, and BCIP/NBT loading.

**Figure 3 diagnostics-09-00203-f003:**
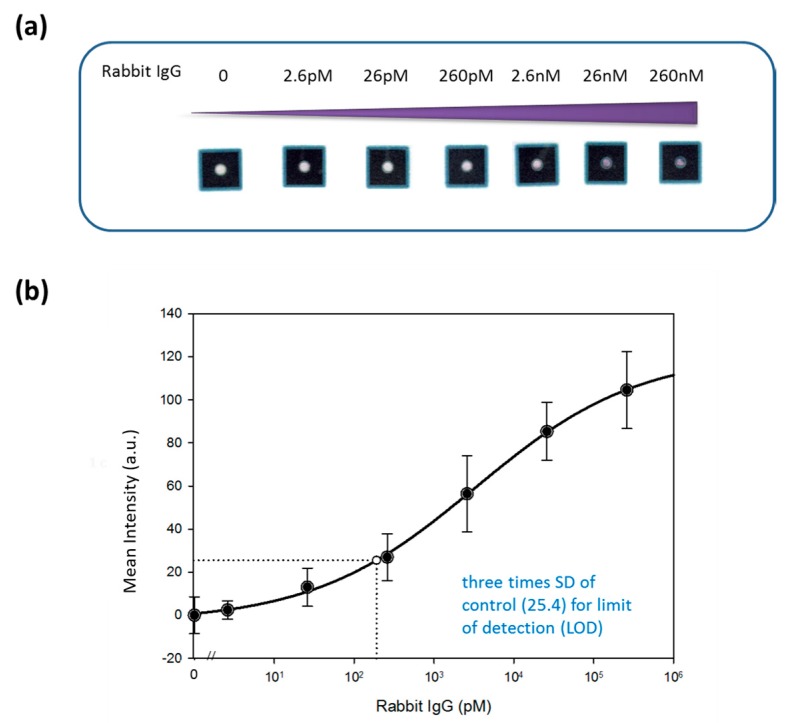
Results of rabbit IgG detection using the O-PAD. (**a**) Colorimetric response. (**b**) Calibration curve of rabbit IgG using mean gray-scale intensity versus several concentrations of rabbit IgG (*n* = 6).

**Figure 4 diagnostics-09-00203-f004:**
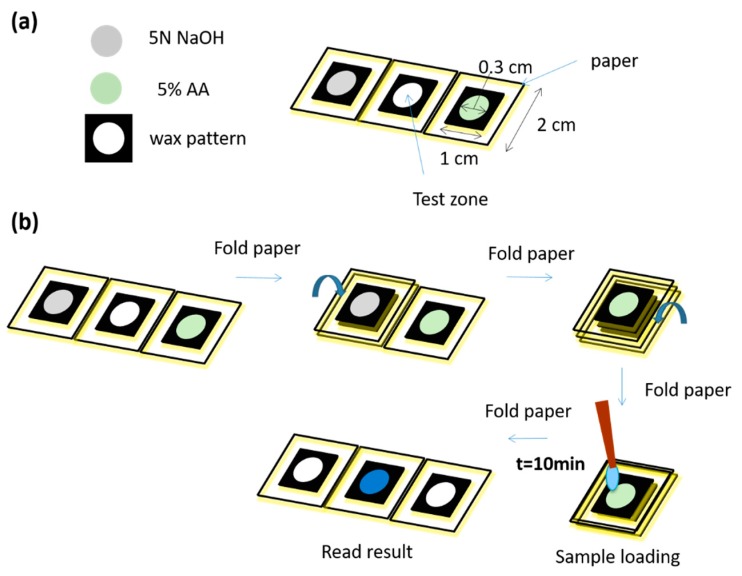
Schematic illustrations of O-PAD for paraquat detection. (**a**) 5 µL of 5 N NaOH and 6 µL of 5% (*w*/*v*) ascorbic acid (AA) were preloaded onto the left and right parts of our PAD, respectively. (**b**) The reagent zones containing NaOH and AA were folded onto the test zone sequentially. 8 µL of test sample was loaded onto the test zone, and the device stood for 10 min at 25 °C. Paraquat, in contact with NaOH and AA, changed to paraquat radical ion (blue color).

**Figure 5 diagnostics-09-00203-f005:**
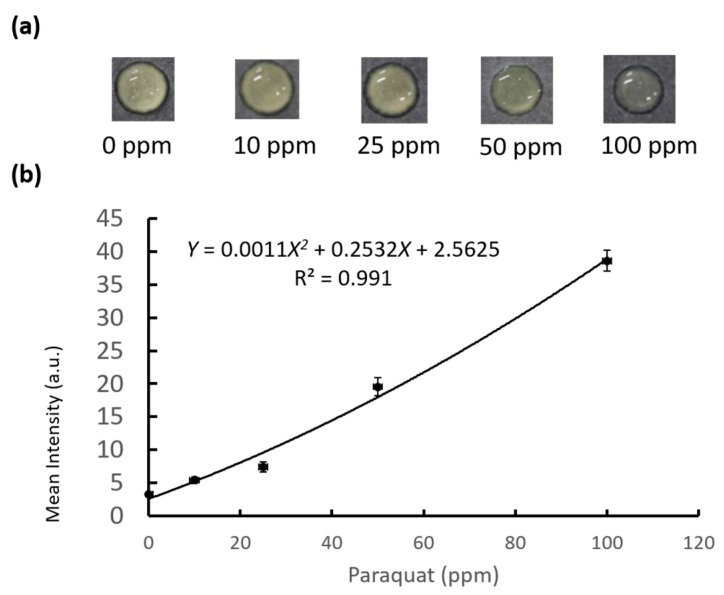
Result of paraquat detection using the O-PAD. (**a**) Colorimetric response. (**b**) The calibration curve of paraquat using mean RGB intensity versus several concentrations of paraquat (*n* = 3).

**Table 1 diagnostics-09-00203-t001:** Comparisons of conventional ELISA and paper-based ELISA (P-ELISA) to origami-based paper devices (some information adapted from reference [3]).

	Conventional	P-ELISA	Origami
Pre-loaded reagents for assay	No	No	Yes
Reagents needed for conducting assay	Multiple reagents	Multiple reagents	One reagent or sample only
Equipment	Multiple tips	Multiple tips	Single tip
Convenience	Low	Low	High
Environment for reactions of detection	Less stable	Less stable	More stable
Required antigen volume	70 μL	3 μL	2 μL
Total complete time	213 min	51 min	46 min
LOD	5.7 pM	18 nM	201 pM

LOD—limit of detection.

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
