# Peer review of "Two Potential Clinical Applications of Origami-Based Paper Devices"

_diagnostics, 2019, doi:10.3390/diagnostics9040203_

Round 1
Reviewer 1 Report
Authors fabricated the paper-based analytic system for detection rIgG and paraquat. O-PADs in this manuscript is very simple and cheap so it can be used for POCT. In addtion this device does not need expensive reading system.
For ELISA system, washing step is very improtant for the removal of non-bind detection antibody since it can generate significant noise level. So normal ELISA require at least 3 times of complete washing for removing the detetion antibody reagent. If not, significant noise will be generate and this noise will disturb the signal to noise ratio. In this manuscript, PAS has extra washing part. However, this washing part connot remove all non-bound detection antibody. Authors should explain how the washing step can work efficiently in their device with detail data.
In addition, authors used direct ELISA to detect rabbit IgG. Normally, direct ELISA shows lower sensitivity than sandwich ELISA. However, authors claimed that their device showed very low LOD. How does it possible? Authors immoblized rIgG for 10 min. Normally, immoblization of IgG on 2D substrate requires more time. Did IgG immobilized completely? Or it will be washed out in washing step.
For detection of paraquat, it would be better to show the colot data like Fig. 3 (a) for better understanding.
Author Response
Reviewer 1:
Authors fabricated the paper-based analytic system for detection rIgG and paraquat. O-PADs in this manuscript is very simple and cheap so it can be used for POCT. In addtion this device does not need expensive reading system.
For ELISA system, washing step is very improtant for the removal of non-bind detection antibody since it can generate significant noise level. So normal ELISA require at least 3 times of complete washing for removing the detetion antibody reagent. If not, significant noise will be generate and this noise will disturb the signal to noise ratio. In this manuscript, PAS has extra washing part. However, this washing part connot remove all non-bound detection antibody. Authors should explain how the washing step can work efficiently in their device with detail data.
Ans: Based on our previous work in 2010,1 the design of paper-based ELISA, and the origami paper-based ELISA in this manuscript, contains top and bottom faces of test zones in paper microzone plates. Different from conventional ELISA, one of the advantages of this design is that we can wash the zone by adding buffer from the top face of the zone and pressing the bottom face against the blotting zone. The washing buffer runs through the zone and carries unbound reagents with it. We also added the clarification in our manuscript: line 265-267
“Paper-based ELISA, for incidence, requires less amount of samples and complete time than conventional ELISA because of their high surface-to-volume ratio of paper fibers in the test zone.3”
For the proper amount and concentration of washing buffer in this study, was have done some testing beforehand. To prevent washing out of analyte and detection antibody, we have tried to optimize the volume and the percentage of Tween-20 of PBS in washing steps. In figure 1A, we tried loading different volume of PBS for the washing step, and found loading 3 µL PBS showed higher intensity of color formation. To optimize the concentration of Tween-20 in PBS, we tried to use 3 µL PBS with different concentrations of Tween-20. We found PBS with 0.01% Tween-20 was suitable for further study (Figure 1B).
(Figure 1: please see the attachment)
In addition, authors used direct ELISA to detect rabbit IgG. Normally, direct ELISA shows lower sensitivity than sandwich ELISA. However, authors claimed that their device showed very low LOD. How does it possible? Authors immoblized rIgG for 10 min. Normally, immoblization of IgG on 2D substrate requires more time. Did IgG immobilized completely? Or it will be washed out in washing step.
Ans: As our previous study showed and discussed, paper-based ELISA needs smaller amount of samples and shorter incubation time.1 It is attributed to the high surface-to-volume ratio of paper fibers in the test zone. For example, in a test zone with 5mm diameter and 0.18mm height, there is approximately 30% volume occupied by paper fibers. This configuration makes paper-based ELISA different from conventional 2D ELISA, which leads to shorter incubation time. The whole process of paper-based ELISA usually could be completed within one hour.
For detection of paraquat, it would be better to show the colot data like Fig. 3 (a) for better understanding.
Ans: Thanks a lot for the suggestion. We have improved our Fig. 5 accordingly.

Reviewer 2 Report
This research manuscript describes the development of an origami-based paper ELISA device and its potential applications in diagnosis of viral infection and food poisoning at point-of-care settings. The authors demonstrated that their origami-based paper device is sensitive in detection of IgG antibodies and paraquat. The manuscript is interesting and well-written. However, there are a few things that the authors need to address.
1. Besides sensitivity, the authors should also demonstrate specificity of their device by testing it with IgA antibodies, IgM antibodies, and other organophosphate poisons such as malathion and parathion.
2. The authors should use their device for real sample testing, for example tear samples spiked with IgG antibodies and blood samples spiked with paraquat.
3. Lines 177-178: “…the LOD for IgG using our device was 201 ρM, which was three times the standard deviation of the signals from our control (25.4).” Please provide a clear description.
4. In order to have a fuller comparison between the conventional and origami-based paper devices, it would be very helpful to include both the required sample volume and required assay time in Table 1.
5. Besides advantages, the authors should also discuss the limitations of their device and suggest potential strategies to overcome them.
Author Response
Reviewer 2:
This research manuscript describes the development of an origami-based paper ELISA device and its potential applications in diagnosis of viral infection and food poisoning at point-of-care settings. The authors demonstrated that their origami-based paper device is sensitive in detection of IgG antibodies and paraquat. The manuscript is interesting and well-written. However, there are a few things that the authors need to address.
Besides sensitivity, the authors should also demonstrate specificity of their device by testing it with IgA antibodies, IgM antibodies, and other organophosphate poisons such as malathion and parathion. The authors should use their device for real sample testing, for example tear samples spiked with IgG antibodies and blood samples spiked with paraquat.
Ans (for 1 and 2): This concern is legitimate, but addressing it, and optimizing the assay (without other inferences), will require (as it did with conventional ELISA) an extensive engineering effort. We are prepared to begin this optimization, but expect it to take another year to complete. Sample testing for either tears or blood is helpful for confirming the results in this manuscript, and we plan to continue sample testing experiments in the near future. However, this process will require Institutional Review Board (IRB) document preparation, review, and certification that would postpone sample testing for at least a few months.
We believe that the current manuscript outlines the method, points to a potential major improvement/modification in a broadly used bioanalytical system, and establishes our focus for improvement. We would argue that its content is appropriate for an initial manuscript.
Lines 177-178: “…the LOD for IgG using our device was 201 ρM, which was three times the standard deviation of the signals from our control (25.4).” Please provide a clear description.
Ans: We determined the limit of detection for IgG using a signal that was three times the standard deviation of the blank sample1. Briefly, we fit the signal calculated from three times the standard deviation of the blank sample (25.4) to the IgG calibration curve from our device, and obtained 201 pM as the LOD of our device. We added additional descriptive text to our revised manuscript (lines 180-182).
In order to have a fuller comparison between the conventional and origami-based paper devices, it would be very helpful to include both the required sample volume and required assay time in Table 1.
Ans: Thank you for this suggestion. The sample volume required for our origami ELISA was 2 µL and completion time was 46 minutes. Conventional ELISA requires 70 µL and 213 minutes to complete. This comparison is provided in Table 1.
Besides advantages, the authors should also discuss the limitations of their device and suggest potential strategies to overcome them.
Ans: We have discussed limitations and potential future strategies at the end of this study as follows:
Conclusions, line 286-291.
“However, there are still some limitations associated with our device. First, while providing advantages over original paper-based ELISA, our device still could not reach the sensitivity of conventional ELISA; this limitation may be improved by optimizing the assay to suppress the background signal. Second, the long-term storage stability of preloaded reagents such as ascorbic acid, which oxidizes if exposed to oxygen, may limit usage. We believe that additional developments, especially those aimed at improving long-term dry-state storage, would improve our device and broaden its applicability.”

Round 2
Reviewer 1 Report
Authors answered according to my comment well. However, most of the answer was from the previous reference. Then, what is the novelty of this manuscript? Authors should emphasis the advantage of developed O-PAD comparing with the previous parper based ELISA. In addition, it would be better to compare the result including the limit of detection in same table, Table 1.
In table 1. the required antigen volume is 70 and 2 iL or μL? Please check again.
In Figure 5 (a), the color of back ground (outer circlular area) is not uniform. Authors modified the image for calculate the intensity?
Comparign with Fig. 3 (b) and Fig 5 (b), title of y-axis is different: mean gray scale intensity and delta intensity. Is the calculating method different?
Author Response
Reviewer 1:
Authors answered according to my comment well. However, most of the answer was from the previous reference. Then, what is the novelty of this manuscript? Authors should emphasis the advantage of developed O-PAD comparing with the previous parper based ELISA. In addition, it would be better to compare the result including the limit of detection in same table, Table 1.
Ans: O-PAD provides two advantages: (1) to reduce test complexity and the number of required reagents through pre-loaded reagents; and, (2) to provide a wet environment for moisture-sensitive analysis, e.g., paraquat detection. The relevant description is added to line 286 to 288. The comparison of O-PAD to paper based ELISA is also provided as revised in Table 1.
In table 1. the required antigen volume is 70 and 2 iL or μL? Please check again.
Ans: The typo is corrected accordingly.
In Figure 5 (a), the color of back ground (outer circlular area) is not uniform. Authors modified the image for calculate the intensity?
Ans: The images were all original. We substrated the signal in the background individually when calculating the intensity. The typos for the paraquat concentration in the photos of the previous version are also corrected.
Comparign with Fig. 3 (b) and Fig 5 (b), title of y-axis is different: mean gray scale intensity and delta intensity. Is the calculating method different?
Ans: In Fig 3 (b), the y-axis should be delta gray scale intensity; in Fig 5 (b), the y-axis should be delta RGB intensity. We subtracted original test zone background values for each detection result to diminish background noise and analytical variation. To avoid confusion, the y-axis is revised as “Mean intensity (a.u)” in both Fig 3(b) and Fig 5 (b). The relevant description is added in our revised manuscript.

Reviewer 2 Report
The quality of the manuscript has been significantly improved. However, I have 1 further comment. Table 1: replace "ìL" with "μL".
Author Response
Reviewer 2:
The quality of the manuscript has been significantly improved. However, I have 1 further comment. Table 1: replace "ìL" with "μL".
Ans: The typo is corrected accordingly.

Round 3
Reviewer 1 Report
AUthors modified the manuscript according to the reviewer's comments. So I agree to publish this article in Diagnostics.